# Carbon Black CB-EDA Nanoparticles in Macrophages: Changes in the Oxidative Stress Pathway and in Apoptosis Signaling

**DOI:** 10.3390/biomedicines11061643

**Published:** 2023-06-06

**Authors:** Joice Margareth de Almeida Rodolpho, Krissia Franco de Godoy, Patricia Brassolatti, Bruna Dias de Lima Fragelli, Luciana Camillo, Cynthia Aparecida de Castro, Marcelo Assis, Carlos Speglich, Elson Longo, Fernanda de Freitas Anibal

**Affiliations:** 1Laboratório de Inflamação e Doenças Infecciosas, Departamento de Morfologia e Patologia, Universidade Federal de São Carlos, São Carlos 13565-905, Brazil; 2Department of Analytical and Physical Chemistry, University Jaume I (UJI), 12006 Castelló, Spain; 3Centro de Desenvolvimento de Materiais Funcionais, Departamento de Química, Universidade Federal de São Carlos, São Carlos 13565-905, Brazil; 4Centro de Pesquisa Leopoldo Américo Miguez de Mello CENPES/Petrobras, Rio de Janeiro 21941-915, Brazil

**Keywords:** carbon black nanoparticles, J774A.1 cells, reactive oxygen species, viability cell, oxidative stress, apoptosis

## Abstract

The influence of black carbon nanoparticles on J774.A1 murine cells was investigated with the objective of exploring the cytotoxicity of black carbon functionalized with ethylenediamine CB-EDA. The results showed that CB-EDA has a cytotoxic profile for J774.A1 macrophages in a time- and dose-dependent manner. When phagocytosed by the macrophage, CB-EDA triggers a mechanism that leads to apoptosis. In this process, there is an increase in oxidative stress pathways due to the activation of nitric oxide and then ROS. This causes an imbalance in redox function and a disruption of membrane integrity that occurs due to high levels of LDH, in addition to favoring the release of the pro-inflammatory cytokines IL-6, IL-12, and tumor necrosis factor (TNF) in an attempt to modulate the cell. However, these stimuli are not sufficient to repair the cell and the level of mitochondrial integrity is affected, causing a decrease in cell viability. This mechanism may be correlated with the activation of the caspasse-3 pathway, which, when compromised, cleaves and induces cells death via apoptosis, either through early or late apoptosis. In view of this, the potential for cell damage was investigated by analyzing the oxidative and inflammatory profile in the macrophage lineage J774.A1 and identifying potential mechanisms and metabolic pathways connected to these processes when cells were exposed to NP CB-EDA for both 24 h and 48 h.

## 1. Introduction

The last few decades have witnessed the thriving development and widespread use of nanoparticles in a variety of industries, including energy, aircraft, agriculture, business, and medical applications [1]. Nanomaterials have a crystalline or amorphous particulate structure at the nanometer scale, and their size-specificity and other properties distinguish them from many traditional materials. The development of new nanostructured materials has led to increased interest in relation to technology, attracting interest from both academic and industrial fields [2].

Carbon black (CB) is one of the most frequently produced and used nanomaterials, in addition to being a nanoparticle that has commercial advantages related to its low cost. Its applications include usage in medicine, catalysis, environmental remediation, oil exploration and reservoir assessments, as well as nanotechnology. Morphologically, CBs are aggregates of spheroidal nanoparticles, typically 10 to 100 nm in diameter [3].

Nanomaterials can be functionalized by adding molecules to their structure, which can be used to change their characteristics as desired. In this sense, CB has been functionalized with ethylenediamine (EDA) to produce a nanoparticle called CB-EDA. CB has low dispersion in polar solvents and, as a strategy to adjust the interactions of the particle surface with the medium and improve its dispersibility and compatibility, it has been covalently bonded to these polymers [4].

The complexity and diversity of carbon nanomaterials in terms of their size, charge, shape, chemical compositions, preparation methods, surface functionalization, and aggregation tendency are factors that influence the cytotoxicity of these nanoparticles [5]. Carbon black nanoparticles are composed primarily of carbon, with a small number of other elements (including hydrogen and oxygen) that by themselves are non-toxic. However, CBs functionalized with other structures are potentially toxic. Given that their surface area can interact with biological systems, they could have detrimental effects [6]. In the case of CB-EDA, the functionalization with EDA could influence the biological activity of the nanoparticle due to the modification of its physicochemical properties, resulting in potential toxicity and unwanted biological effects [7].

Despite the enormous contributions made by nanotechnology to society, there is great concern about the toxicity of nanomaterials in a wide range of living beings and the environment. To assess the risks involved in the production, use, and disposal of nanomaterials and products containing nanoparticles, there must be an understanding of their hazards and an assessment of human exposure levels [8]. In this context of environmental and human protection and safety, the field of nanotoxicology has emerged. This field deals with the deleterious effects of nanomaterials or nanoproducts on living organisms throughout their life cycle, with the objective of achieving reliable safety assessments and the adequate regulation of the production, use, and deposition of nanomaterials [1]

Due to the scarcity of studies using CB-EDA, there is still no complete understanding of its toxic potential, which makes the exploration of this nanoparticle’s performance extremely important. Relevant tests may include in vitro and in vivo studies that aim to provide information on the connections between the formation of reactive oxygen species (ROS) and reactive nitrogen species (RNS) and the synthesis and release of pro-inflammatory chemicals, as well as cell viability analyses after exposure to nanoparticles [5,9].

Several nanoparticles have gained notoriety in the biomedical field due to their increasing applications with the potential for scientific and technological innovation. These materials are excellent candidates for interfaces with neural systems [10], the detection of tumor cells with greater sensitivity [11], gene therapy [12], and vaccine administration [13], among others.

The cell lines used in this study were employed because lung carcinogenesis develops in epithelial cells, and macrophages are cells that identify small particles to trigger inflammatory responses in the respiratory system [14].

In view of this, the objective of this study was to elucidate the cytotoxic effects of modified CB-EDA nanoparticles in in vitro models by evaluating the cytotoxic profile of CB-EDA in the J774A.1 strain of murine macrophages in the acute phase of exposure.

## 2. Material and Methods

### 2.1. Nanoparticles

CB-EDA nanoparticles were provided by the petroleum industry, specifically from the Centro de Pesquisa Leopoldo Américo Miguez de Mello CENPES/Petrobras, Rio de Janeiro, RJ, Brazil (PETROBRAS).

### 2.2. Characterizations

The samples were characterized via transmission electron microscopy (TEM) and high-resolution TEM (HR-TEM) using a Jeol 2100F microscope operating at 200 kV (Oxford). Particle size was obtained by counting 200 particles using Image J 1.53a software. For the analysis of zeta potential and dynamic light scattering (DLS), a Malvern spectrometer (Marvern Instruments) with Nano-ZS equipment was used.

The increase in particle size, observed via DLS during particle incubation with culture media, was used to assess corona development. The particles were first treated with RPMI medium for 24 and 48 h. They underwent DLS analysis after centrifugation. Following these analyses, the same particles were cleaned three times in distilled water before undergoing another DLS analysis.

### 2.3. Cell Culture

J774A.1 cells were supplied by the Banco de Células do Rio de Janeiro (BCRJ), code:0121, cultured in RPMI medium (SIGMA-ALDRICH, Spring, UT, USA), supplemented with 10% fetal bovine serum (LGC Biotechnology), and maintained in a humidified incubator containing 5% CO_2_ at 37 °C. The concentrations used for in vitro assays using CB-EDA nanoparticles were 1, 50, 250, 500, and 1000 μg/mL at 24 and 48 h.

### 2.4. Cytotoxicity Assay with MTT Salt and EC_50_

The colorimetric method with MTT (Tetrazolium 3-(4,5-dimethylthiazol-2-yl)-2,5-diphenylbromide) salt (SIGMA-ALDRICH, USA) [10] was performed to evaluate cell viability through mitochondrial function integrity, observed by measuring the formation of formazan crystals by mitochondrial enzymes. The larger the assembly of those crystals, the greater the cell viability. First, 1 × 10^4^ cells per well were inoculated into a 96-well flat-bottomed, culture-treated, lidded microplate. After 24 h of adhesion, cells were exposed to CB-EDA NPs at various concentrations. After exposure times of 24 and 48 h, the wells were washed twice with 1× PBS (phosphate buffer saline, pH 7.4) and 200 μL of 0.5 mg/mL MTT solution in 1× PBS and DMEM medium, which was incomplete and without phenol, was added at a ratio of 1:5. The reaction occurred for 4 h at 37 °C and 5% CO_2_. Blank controls were made containing only MTT solution. Then, the chemical agent solution was removed and 100 μL of DMSO agent was added per well, followed by the reading of the absorbance at 570 nm in a photometer (Thermo Scientific ™ Multiskan ™ GO Microplate Spectrophotometer). Cell death control (CTRL+) was performed with 5% Extran enzymatic detergent. The percentage of cytotoxicity was observed by comparing the data obtained with the CTRL-group according to the equation described below. Calculations of the EC50 (the half maximal effective concentration) and the cell viability (% of cytotoxicity = absorbance of the test groups × 100/absorbance of the control group) were carried out using the absorbance measurements. The percentage of cell viability was determined by comparing the data from the management cluster using the following equation:% cytotoxicity = experimental group × 100average of C-group

### 2.5. Cell Viability Analysis with the Neutral Red Dye Assay

The neutral red dye assay was used for the J774A.1 strain in order to confirm cell viability. This method is based on the accumulation of dye in lysosome membranes. The more viable cells, the greater the diffusion of dye through the membrane [11]. In a 96-well flat-bottomed microplate, with treatment for cultivation, 1 × 10^4^ cells were inoculated per well. After 24 h of adhesion to CB-EDA NPs, they were exposed to different concentrations of CB-EDA NPs for 24 and 48 h. Following the different durations of exposure, 100 L of neutral red dye at 30 g/mL in DMEM medium supplemented with 1% SBF and without phenols were added. The wells were then rinsed twice with 1× PBS. The reaction occurred for 2 h at 37 °C and 5% CO_2_. Then, the reagent solution was removed and 200 μL of the diluent in each well received the addition of ethanol at 50% with 1% acetic acid before a spectro-reading and photometry of the absorbance at 540 nm. (Multiskan GO Microplate Spectrophotometer from Thermo Scientific (Waltham, MA, USA)). The formula presented in the previous section was used to calculate the percentage of cytotoxicity.

### 2.6. LDH (Lactate Dehydrogenase)

By evaluating LDH production in accordance with the protocols of the CyQuantTM LDH Cytotoxicity Assay Kit, cell membrane damage caused by cytotoxicity was determined (Invitrogen). First, 1 × 10^4^ cells were planted into each well of a 96-well plate. Varying concentrations of CB-EDA NPs were added after the 24 h period recommended for cell attachment. The supernatant from each group was collected after the two exposure times (24 and 48 h) and 50 μL was added to a fresh 96-well plate. Each well received 50 μL of the reagent solution for 30 min. Thermo Scientific’s Multiskan GO Micro-plate Spectrophotometer was used to measure the absorbance at 490 and 680 nm, which were then subtracted to determine the percentages of cytotoxicity. The results obtained at 680 nm from 490 nm were subtracted to determine LDH activity, and the formula below was used to compute the percentage of cytotoxicity: The greatest LDH activity was exhibited by 10 μL of lysis solution, whereas the wells with spontaneous LDH activity contained 10 μL of water.
% cytotoxicity = (Experimental group spontaneous LDH activity) × 100 (Maximum LDH activity spontaneous LDH activity)

### 2.7. Clonogenic Assay

The ability of J774A.1 cells to recover after being exposed to CB-EDA NPs at concentrations of 1, 50, 250, 500, and 1000 μg/mL after 24 and 48 h of exposure was determined based on the ability of cells to form colonies via a clonogenic assay [12]. First, 1 × 10^2^ cells per well were inoculated into a 6-well flat-bottomed microplate with treatment for cultivation and with a lid. After 24 h of adhesion, the cells were exposed to different concentrations of CB-EDA NPs for 24 and 48 h. A control group was also made (cells + complete medium). After the exposure times, the supernatant was discarded and a new complete culture medium was added. After 7 days of recovery, and medium was exchange twice and cells were fixed with methyl alcohol P.A. (absolute methanol) and stained with 0.1% crystal violet (diluted in distilled water). After dilution with SDS, the colonies were counted and the wells were imaged. Colony counts were performed after taking pictures of the wells and using the ImageJ 1.53a program (Schindelin J. et al., 2012) [15]. A Thermo Scientific ™ Multiskan ™ GO Microplate Spectrophotometer was used to measure the absorbance at 570 nm.

### 2.8. Cell Morphology

The cell morphologies of the J774A.1 strain after exposure to CB-EDA NPs at different concentrations and of the control group after 24 and 48 h of exposure were observed via light microscopy. In this process, 1 × 10^4^ cells per well were inoculated into a 96-well flat-bottomed microplate and, after 24 h of adhesion, differing CB-EDA NP concentrations were exposed to cells at different concentrations after 24 and 48 h of exposure. After the two exposure times, the wells were observed in an Axiovert 40 CFL optical microscope (Zeiss) with a 10× objective lens, with images captured using the coupled camera model LOD-3000 (Bio Focus) and analyzed using Future WinJoeTM software version 2.0 at the high-resolution limit of 100×.

### 2.9. Quantification of IL-6, IL-12, and TNF Cytokines

For quantification, direct ELISA kits for IL-6, IL-12, and TNF cytokines (OptEIATM Kit, BDBiosciences) were used, following the protocol described below. For this procedure, 96-well high-affinity microtiter plates were used. Between each step, plates were washed with 300 μL/well of wash solution (1× PBS + 0.05% Tween 20). After the sensitization of the plates with 100 μL/well of the specific capture antibody in carbonate and/or phosphate buffer, incubation was carried out for 16 h at 20 °C. Then, reaction blocking was performed with 200 μL/well with 10% (non-fat) milk, followed by incubation for 1 h at room temperature. After that, 50 μL of samples and the titration curve of the cytokine standards were applied with a 2 h incubation, which was also carried out at room temperature. The detection antibody conjugated to the peroxidase enzyme was added at 100 μL/well for 1 h and 30 min, protected from light. Then, 100 μL/well of TMB substrate was applied and the plates were incubated, still protected from light, for approximately 15 to 30 min. The reaction was blocked with the application of 50 μL/well of 2N sulfuric acid and plates were read at the 450 nm wavelength using a plate spectrophotometer (Thermo Scientific ™ Multiskan ™ GO Microplate Spectrophotometer). Concentrations were calculated using cytokine titration curve standards and the final concentrations were expressed in pg/mL.

### 2.10. Detection of Reactive Nitrogen Species (RNS) Production via the Griess Reaction

The detection of RNS was quantified using the Griess reaction, which measures NO production indirectly via nitrite ion (NO_2_^−^) measurements. In this process, 1 × 10^4^ cells per well were inoculated into a 96-well flat-bottomed microplate with a lid. After 24 h of cell adhesion, J774A.1 cells were exposed to different concentrations of CB-EDA NPs for 24 and 48 h. Following the exposure times, 50 μL of the supernatants were taken and put on a fresh plate. Next, 50 μL of the Griess solution (a 1:1 mixture of solution A [1% sulfanilamide in 5% phosphoric acid] and solution B [0.1% N-1-naphthylethylenediamine dihydrochloride]) was put on the plate and left for 15 min at room temperature. A Thermo Scientific ™ Multiskan ™ GO Microplate Spectrophotometer was used to measure the absorbance at 540 nm. A standard curve with known values of nitrite in mM was used to measure the nitrite concentration in the supernatant (200–1.5 mM).

### 2.11. Detection of Reactive Oxygen Species (ROS)

The fluorescent probe DCFH-DA (2′,7′-Dichlorodihydrofluorescein Diacetate) was used to measure the generation of ROS (Sigma-Aldrich). 1 × 10^4^ cells/well were placed on a 96-well plate with a maximum capacity. Concentrations of CB-EDA NPs were applied every twenty-four hours to promote cell adherence. After the two exposure times (24 and 48 h), each well received a half-hour application of the DCFH-DA probe, diluted in medium. The wells were cleaned with PBS 1×, and a Spectra Max i3 ^®^ plate photometer was used to measure the visible light emissions between 485 and 530 nm (Molecular Devices). The results were converted into percentages using an equivalent formula that was previously discussed in relation to the MTT test.

### 2.12. Quantification of Mitochondrial Electrical Potential

A specific fluorochrome called rhodamine 123 was used to mark mitochondria in live cells. Since it is a cationic fluorochrome (green in color), it is drawn to the mitochondrial membrane’s strong negative electrical potential and incorporates itself there. The rise in cytosolic green fluorescence, which denotes the diffusion of rhodamine 123 from the mitochondria to the cytosol, can be used to identify changes in mitochondrial integrity (Ronot, X. et al., 1986) [16]. In this process, 1 × 10^5^ cells were put into each well of a flat-bottomed 24-well plate that had been treated for cultivation and was equipped with a lid. Different concentrations of CB-EDA NPs were applied after 24 h of adhesion. The plates were centrifuged, cleaned with 1× PBS following exposure times of 24 and 48 h, and then 100 μL/well of rhodamine 123 antibodies (5 mg/mL diluted in ethanol) was added. The reaction occurred for 30 min at 37 °C and with 5% CO_2_, protected from light. Then, the cells were removed with a scraper and resuspended in microtubes with 300 μL of 1× PBS. Readings were performed on an Accuri ™ C6 BD Biosciences flow cytometer, selecting a gate with 10,000 events. Analysis was performed using FlowJo ™ version XV software (BD Biosciences).

### 2.13. Quantification of Caspase 3 Levels

For apoptosis detection via the determination of increased caspase-3 activity, the EnzChek^®^ Caspase-3 Assay Kit #1 was used to continuously monitor the activity of caspase-3 and closely related proteases in cell extracts, via a substrate derived from 7-amino-4-Methylcoumarin Z-DEVD-AMC, which produces a bright blue fluorescent product after proteolytic cleavage. A 24-well flat-bottomed plate with a lid was inoculated at 1 × 10^5^ cells per well. After 24 h of adhesion, the different concentrations of CB-EDA NPs were exposed to J774A.1 cells and the control group at 24 and 48 h. After the estimated times, the plate was centrifuged and the wells were scraped with 1× PBS. The cell extract was lysed with lysis buffer for 30 min at −20 °C and centrifuged at 5000× *g* for 5 min. Then, 50 μL of the supernatant was added to a new 96-well microplate and 50 μL of the reagent solution (Z-DEVD-AMC in reaction buffer) was added. The reaction occurred for half an hour at this temperature. The light emissions were scanned at 342 nm excitation and 441 nm emissions within the Spectra MAX i3 ™ equipment (Molecular Devices).

### 2.14. Analysis of Cell Death Due to Apoptosis/Necrosis Using the Annexin V Marker

The BD Biosciences detection kit was used to measure the amount of cell death caused by early and late apoptosis using the annexin V marker. In this process, 1 × 10^5^ cells per well were grown in a 24-well plate. Concentrations of CB-EDA NPs were added to cells after 24 h for cell adhesion. The plates were centrifuged and cleaned after the two different exposure periods (24 and 48 h). Then, additional Annexin V and 7AAd antibodies were added (1 μL/well in binding buffer). The response persisted for 24 h; associate degree analysis was performed using a flow cytometer from AccuriTM C6 baccalaureate Biosciences (gate: 10,000 events). Information analysis was developed using the FlowJo XV package (BD Biosciences) 2.15 for statistical analysis.

The results were analyzed using GraphPad Prism 7.0 (San Diego, CA, USA) and Sigma plot package (version 14). All studies were performed in biological triplicates. The identification of discrepant information was performed through Grubbs analysis, followed by analysis of the distribution of variables in order to test the normality (Shapiro–Wilk test) and equal variance (Levene method). For the analysis of multiple comparisons, two-way multivariate analysis with Tukey logical fallacy tests were used to evaluate the variance between groups in terms of constant-quantity information (results are presented as means and standard deviations) and statistical information. The Kruskal–Wallis test was used, along with Dunn’s test. Statistical significance was established at *p* < 0.05.

## 3. Results

Figure 1A,B shows the scanning electron microscopy (SEM) and transmission electron microscopy (TEM) images of CB nanoparticles. Good granulometric homogeneity of the particles was observed, with their unitary diameters varying between 10 and 40 nm. The particles showed a highly aggregated irregular spheroid morphology. Figure 1C shows high-resolution TEM (HR-TEM) images of the CB nanoparticles, where it is possible to observe the concentric patterns of carbon layers aligned with the inner circumference of the particle [13].

The behavior of these nanoparticles in solution (water and culture medium) was analyzed through the measurements of zeta potential and DLS, since the interaction between the surface of the particle and the molecule of the medium affects the ionization of the surface and therefore the stability of the particles’ dispersion [14]. The zeta potential values observed in this study were −4.24 ± 1.21 and −3.69 ± 0.82 mV when dispersed in water and culture medium, respectively, showing that the medium did not significantly change the surface charge of the nanoparticles. Figure 1D shows the average particle size and standard deviations values obtained through the DLS technique. In water, we observed that at 24 and 48 h, the nanoparticles formed agglomerates of approximately 750 nm. When these CB nanoparticles were incubated with the culture medium for 24 h and 48 h, a drastic increase in their size was observed, reaching 2987 ± 108 and 3375 ± 119 nm in 24 h and 48 h, respectively. After washing and centrifugation, although a small reduction in the size of these agglomerates was observed, they were still very large and stable. This considerable increase in agglomerates in the culture medium occurred due to the bonds that the proteins contained in the culture medium made between the surfaces of the nanoparticles, thus forming increasingly larger agglomerates [9,17].

The MTT assay was performed to assess the metabolic activity of J774.A1 macrophages exposed to various NP CB-EDA concentrations (1, 50, 250, 500, and 1000 μg/mL) at 24 and 48 h. CB-EDA NPs showed cytotoxic effects on macrophage viability (Figure 2A) at 24 and 48 h, with significant increases observed at concentrations of 250, 500, and 1000 μg/mL when compared to control groups. EC_50_ values were calculated according to each concentration and exposure time, 24 h and 48 h, as shown in Figure 2 C, D. At 24 h, the EC50 was 12.9 μg/mL and at 48 h it was 27 μg/mL.

In Figure 2B, the results of cell viability analysis using the neutral red dye assay are presented. It can be observed that there was a significant reduction in the number of viable cells in the J774A.1 lineage only at the concentration of 1000 μg/mL when compared to the control group, both at 24 h and at 48 h.

The LDH assay data show that there was a significant difference at 24 h and 48 h in the group exposed to 1000 μg/mL when compared to the control group (Figure 2E).

The J774.A1 macrophage colony formation test (clonogenic assay) was carried out to demonstrate the decrease in cell viability in the control group and in groups treated with CB-EDA NPs with doses of 1, 50, 250, 500, and 1000 μg/mL at 24 and 48 h (Figure 3). When compared to the control group, the numbers of colonies decreased after 24 h with doses of 250, 500, and 1000 μg/mL of CB-EDA NPs. When compared to the control group, the numbers of colonies with 50, 250, 500, and 1000 μg/mL of CB-EDA NPs decreased after 48 h (Figure 3A).

The wells’ photographic patterns were followed during the measurements by observing the wells’ absorbance. Figure 3B,C shows that there was a substantial decrease in the number of colonies in the groups exposed to 50, 250, 500, and 1000 μg/mL of CB-EDA NPs after 48 h and in the groups exposed to 250, 500, and 1000 μg/mL of CB-EDA NPs after 24 h as compared to the control group.

Figure 4 presents qualitative data to demonstrate the cellular morphologies of J774.A1 macrophages after 24 and 48 h of exposure to CB-EDA NPs at different concentrations (1, 50, 250, 500, and 1000 μg/mL) and in the control group. It was possible to observe decreases in size and changes in the shape of cells at 24 h in the groups exposed to 50, 250, 500, and 1000 μg/mL when compared to the control group (Figure 4). At 48 h, this decrease was also observed in groups exposed to 1, 50, 250, 500, and 1000 μg/mL when compared to the control group (Figure 4).

The expression of TNF cytokines (mono/mono), IL-6, and IL-12 are shown in Figure 5. In the 48 h period of exposure to CB-EDA NPs, significantly higher production of the cytokines TNF and IL-6 was observed in the group exposed to a concentration of 1000 μg/mL when compared to the control group (Figure 5A,B). No significant detection differences in levels were observed at 24 h for TNF or IL-6. IL-12 expression was not significantly different at the estimated times, 24 and 48 h, when compared to the control group (Figure 5C).

The results of the quantification of reactive nitrogen species (RNS), performed in J774A.1 cells at 24 and 48 h at different concentrations of CB-EDA NPs, are shown in Figure 6. There was a significant increase at 24 and 48 h in cells exposed to concentrations of 500 and 1000 μg/mL when compared to the control group (Figure 6).

After exposure to CB-EDA nanoparticles at doses of 1, 50, 250, 500, and 1000 μg/mL for 24 and 48 h, the J774.A1 macrophage generation of reactive oxygen species (ROS) was examined (Figure 6). When compared to the control group, the rise in ROS at doses of 250, 500, and 1000 μg/mL increased significantly after 24 and 48 h (Figure 6B).

The electrical potential of the mitochondrial membrane (ΔΨm) was analyzed in J774A.1 cells exposed to different concentrations (1, 50, 250, 500, and 1000 μg/mL) of CB-EDA NPs and at 24 and 48 h. The electrical potential is essential to maintaining the physiological function of the respiratory chain and generate ATP. When compared to the control group, a substantial decline was observed at 24 h in cells exposed to CB-EDA NP concentrations of 50, 250, 500, and 1000 μg/mL. When compared to the control group, the levels observed for cells exposed to 250, 500, and 1000 μg/mL of CB-EDA NPs significantly decreased after 48 h (Figure 6C).

Figure 6D shows the results of caspase-3 measurements when cells were exposed to CB-EDA NPs for 24 h. There was a significant increase in the production of caspase 3 in groups exposed to 50, 250, and 500 μg/mL concentrations when compared to the control group. At 48 h, we observed a significant increase in cells exposed to concentrations of 500 and 1000 μg/mL of CB-EDA NPs when compared to the control group.

Figure 7 shows illustrative data from dot plots determined based on flow cytometry, demonstrating early and late apoptosis. The cell exposure times to CB-EDA NPs were 24 and 48 h, with concentrations of 1, 50, 250, 500, and 1000 μg/mL. The results were compared to the control group, and are expressed as percentages of fluorescence; 7AAD (FL-3) on the *X* axis represents late apoptosis and Annexin V/PE (FL-2) on the *Y* axis represents early apoptosis.

In 24 h, there was a significant increase according to the percentage of fluorescence both in early and late apoptotic cells exposed to concentrations of 50, 250, 500, and 1000 μg/mL of CB-EDA NPs when compared to the control group (Figure 8A–D). In 48 h, there was a significant increase in initial apoptotic cells in groups exposed to 50 and 250 μg/mL of CB-EDA NPs when compared to the control group. In late apoptotic cells, there was a significant increase upon exposure to 250, 500, and 1000 μg/mL of CB-EDA NPs when compared to the control group (Figure 8A–D).

The heatmap graph (Figure 8B,C) displays the average number of early and late apoptotic cells in the J774.A1 cell line at the 24 and 48 h marks. The graphs are colored; blue represents the minimum value and red represents the maximum value calculated based on the averages of the numbers of cells.

## 4. Discussion

To demonstrate the cytotoxic profile of CB-EDA NPs in the J774.A1 macrophage cell line at 24 and 48 h, a series of studies based on cytotoxicity were performed, observing the ability of this nanoparticle to stimulate oxidative stress pathways (RNS and ROS) causing damage to cell membrane integrity and leading to reduced cell viability. Exposure to different NP concentrations activated a pro-inflammatory response in the form of TNF, IL-6, and IL-12 and, consequently, the caspase-3 pathway, initiating an apoptotic cellular response.

The biological behavior of nanoparticles is determined by their characteristics, as their size can interfere with how they act or do not act on the cell surface, exposing them to cytotoxic risks. Many authors have reported that toxicity may be related to the size of the material associated with nanoparticles and a greater understanding these interactions and their toxicological impact is essential [18,19,20].

NP and protein corona interactions are crucial to this understanding, because these interactions are related to the mechanism by which NPs enter or do not enter cells. Protein corona formation causes NPs to lose their targeting ability and means that they can be easily phagocytosed by macrophages [21]. However, the direct relationships between cells and their resulting effects on NPs must be studied in detail [22].

As widely used assays for measurements of cellular metabolic activity, the MTT and neutral red dye assays have been found to be indicative of cellular redox activity when a decrease in cell viability occurs during the exposure of cells to nanoparticles [23]. The decrease in viability caused by CB-EDA NPs at 250, 500, and 1000 μg/mL which we observed in macrophages corroborates the results of studies that used A549 (human lung adenocarcinoma lineage) and 3T3 (murine fibroblast lineage) cells with exposure to carbon black (CB). These results showed that in 24 h there was a reduction in cell viability at the highest concentrations when compared to the control groups [24,25]. Another study demonstrated a decrease in cell viability at 24 and 48 h when LA-9 cells were exposed to CB-EDA NPs [9]. To confirm the cytotoxicity indicated by the MTT and neutral red dye assays, we used the colony formation assay technique, which allowed us to quantify cell survival without any type of staining [26,27]. This assay confirmed the results of the MTT assays in regard to the concentrations of 250 μg/mL and 1000 μg/mL of CB-EDA NPs at 24 and 48 h, with reductions observed in both the number of colonies and the quantifications obtained.

It should be noted that this decrease in macrophage viability may be related to the permeability of the plasma membrane, since LDH levels were increased. In this way, when the nanoparticle comes into contact with cells, a rapid release of lactate dehydrogenase enzymes can occur, causing a change in plasma membrane permeability, thus leading to an estimation of cytotoxicity [28,29,30]. In this sense, single-walled carbon nanotubes and graphene oxide also induced a concentration-dependent increase in LDH in murine peritoneal macrophages [31].

The ability of nanoparticles to penetrate cells can generate an increase in the production of the oxidative stress pathway (ROS), which plays a fundamental role in the control of several biological processes, as it easily crosses cellular barriers, altering the overall integrity of the cell. The data obtained in this study at 24 and 48 h of exposure to CB-EDA NPs in J744.1 macrophages suggested the increased production of ROS at higher concentrations of exposure. This increase may be linked to the redox regulation that occurs in cells, that is, the balance between the formation and elimination of ROS, destroying the microenvironment, which can lead to cell death [32,33]. Our data corroborate those of Godoy et al., 2022 [21], who described an increase in ROS production at the highest concentrations of carbon nanotubes (ONCT-TEPA), in tests also performed on J774.A1 murine macrophages and murine fibroblasts [34]. The literature indicates that when ROS production is exacerbated and exceeds the limit of the cells’ antioxidant defense capacity, proteins and lipids are totally damaged, which can lead to a pathophysiological condition, an inflammatory process resulting in apoptosis [35].

In addition, from the moment that an NP crosses the cell membrane, an inflammatory response can occur, as it leads to the activation of conformational proteins that transmit signals to receptors. The production levels of TNF, IL-6, and IL12 were analyzed as indicators of inflammatory responses. TNF production was significantly higher only after 48 h of exposure to CB-EDA NPs. This corroborates the results of a study by Usman et al., 2020, in which no significant increase in TNF was observed in macrophages exposed to positively charged carbon over 24 h or in hepatocytes [24]. These pro-inflammatory cytokines play a key role in macrophage maturation and the regulation of anti-inflammatory properties [36,37].

In the inflammatory process that occurs after NP phagocytosis in the intrinsic cellular environment, cells are susceptible to damage that usually signals the mechanisms of apoptosis [38]. Our findings demonstrate that when macrophages were exposed to CB-EDA NPs for 24 and 48 h, the presence of late apoptotic cells was more frequently observed than that of early apoptotic cells. Interestingly, this death mechanism is related to the reduction of cell content, the fragmentation of internucleosomal DNA, and membrane vacuoles and the formation of small vesicles, which also corresponds to the findings of previous analyses [16,39].

The apoptosis mechanism is usually triggered by the cell itself when it detects damage, and it is therefore called an intrinsic pathway. In addition, the process of apoptosis can also be considered an extrinsic pathway when there is an interaction between the cell and the immune system [40]. Both intrinsic and extrinsic pathways can activate the effector caspase pathways 3, 6, and 7. These pathways, when altered, cleave multiple vital cell components, leading to cell death due to apoptosis [41,42].

Our data demonstrated an increase in caspase activation at higher concentrations, which may have led J774.A1 cells to undergo apoptosis. In addition, apoptosis events can be evaluated initially, presenting the discontinuous condensation of chromatin around the nuclear periphery and via exposure on its surface to phosphatidylserine (PS), whereas viable cells maintain these phospholipids on the inner surface of the plasma membrane. Phosphatidylserine exposure is one of the first events that occur on the cell surface when the cell is undergoing apoptosis [43,44]. Cellular events during apoptosis are associated with cell death (caspase activation, oxidative stress, and cytoplasmic changes).

## 5. Conclusions

The findings of this study show that when J774.1 macrophages were exposed to CB-EDA NPs at concentrations of 500 and 1000 μg/mL, the apoptosis mechanism was triggered via caspase 3, with an increase in the inflammatory response and oxidative stress within the cells. Therefore, such biological facts demonstrate that NPs have cytotoxic potential for this cell type and that high concentrations are harmful to the biological environment, requiring caution in their use in products or services aimed at human health applications.

## Figures and Tables

**Figure 1 biomedicines-11-01643-f001:**
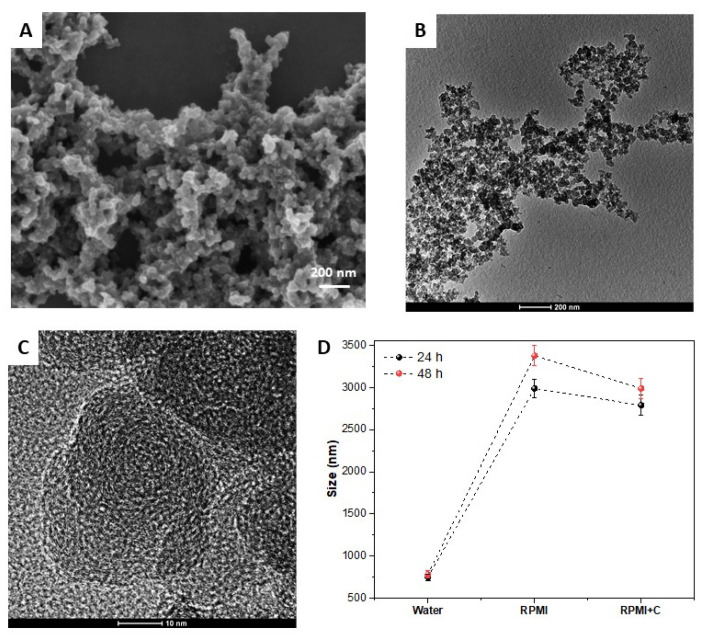
(**A**) SEM and (**B**,**C**) TEM images of the CB-EDA sample. (**D**) Particle size variations, observed via DLS after incubation (24 and 48 h) with the culture medium (RMPI) and after washing with distilled water (RMPI + C).

**Figure 2 biomedicines-11-01643-f002:**
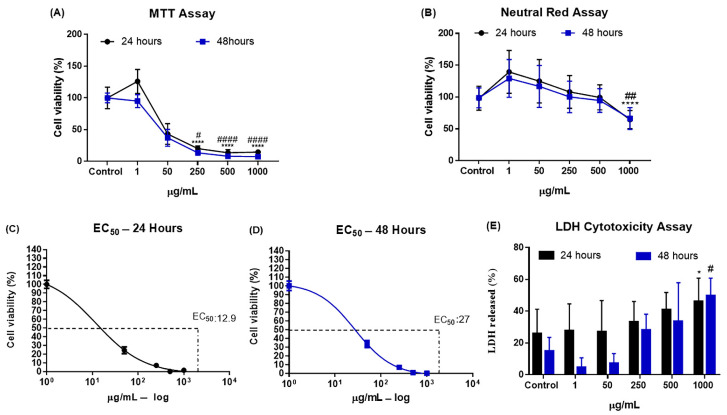
Cell viability as a percentage, determined via MTT and neutral red assays, EC_50_, and LDH. Quantitative data analysis: (**A**) Cell viability as a percentage, determined via MTT assay. (**B**) Cell viability as a percentage, determined via neutral red dye assay. (**C**,**D**) EC_50_. (**E**) LDH assay results. Data are shown after 24 (black) and 48 h (blue) for the control group and the groups exposed to CB-EDA NP concentrations of 1, 50, 250, 500, and 1000 μg/mL. (*) vs. Control after 24 h: * *p* < 0.05; **** *p* < 0.0001. (#) vs. Control after 48 h ^#^
*p* < 0.05; ^##^
*p* < 0.01; ^####^
*p* < 0.0001.

**Figure 3 biomedicines-11-01643-f003:**
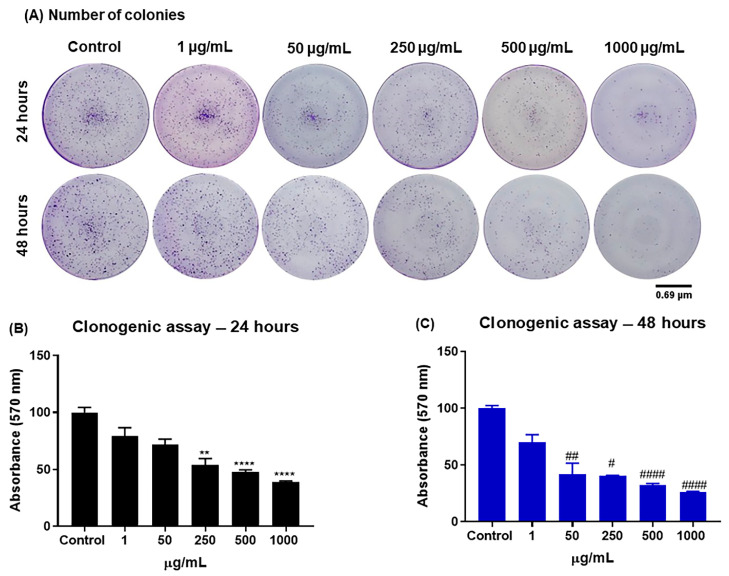
Cell recovery through colony formation. (**A**) Colony formation of J774A.1 macrophage cells after 7 days of recovery and after 24 (black) and 48 h (blue) of exposure to CB-EDA NPs. (**B**,**C**) Absorbance quantification of recovered colonies represented as percentages. ** *p* < 0.01 and **** *p* < 0.0001 vs. Control after 24 h. ^#^
*p* < 0.05, ^##^
*p* < 0.01 and ^####^
*p* < 0.0001 vs. Control after 48 h.

**Figure 4 biomedicines-11-01643-f004:**
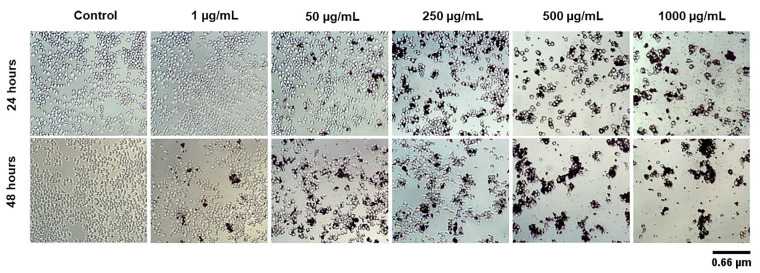
Cell morphologies of J774A.1 cells. Optical microscopy images with 100× magnification, showing the morphology of cells exposed to CB-EDA NPs concentrations of 1, 50, 250, 500, and 1000 μg/mL and the control group: 24–48 h. Qualitative data analysis.

**Figure 5 biomedicines-11-01643-f005:**
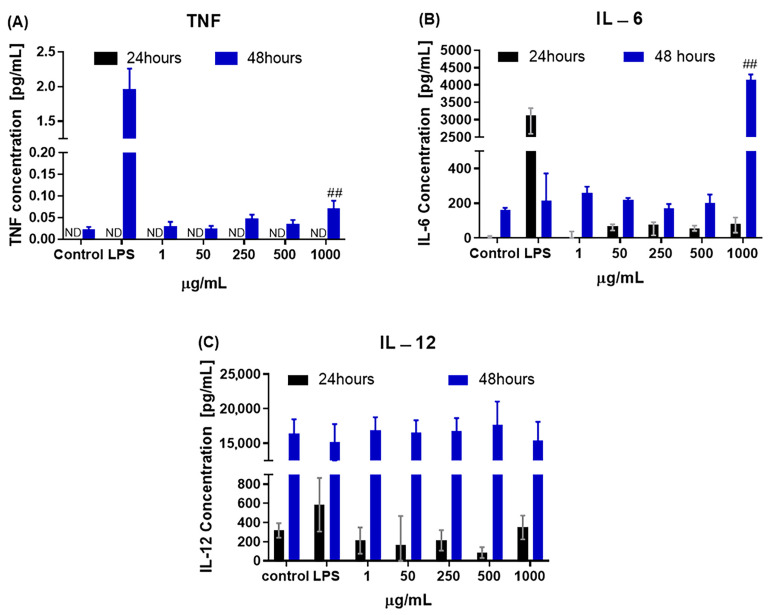
Pro-inflammatory markers TNF, IL-6, and IL-12. (**A**) Detected TNF, (**B**) IL-6, and (**C**) IL-12 levels after 24 (black) and 48 h (blue) of exposure to CB-EDA NPs. ^##^
*p* < 0.01. Control after 48 h. ND (not detected).

**Figure 6 biomedicines-11-01643-f006:**
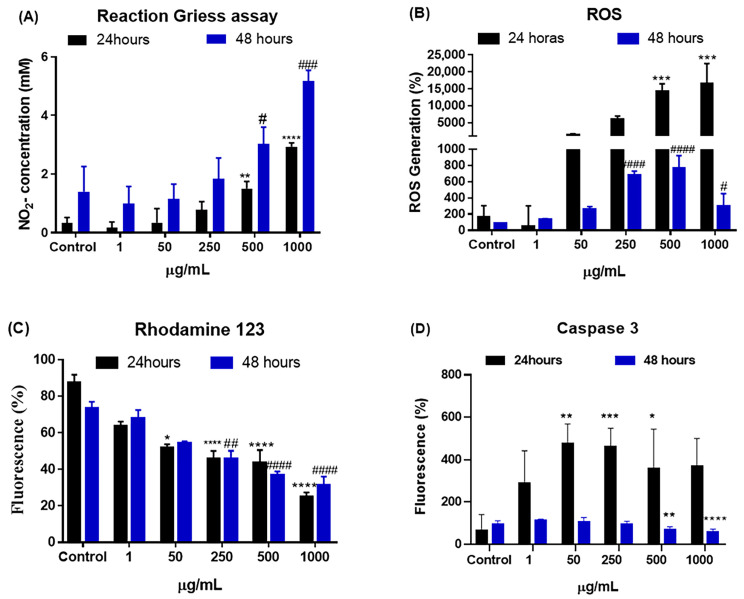
RNS/ROS oxidative stress, electrical potential of the mitochondrial membrane, and caspase-3 activity: (**A**) Detected RNS, (**B**) ROS, and (**C**) electrical potential of the mitochondrial membrane and (**D**) caspase-3 activity after 24 h (black) and 48 h (blue) of exposure to CB-EDA NPs. * *p* < 0.05, ** *p* < 0.01, *** *p* < 0.001 and **** *p* < 0.0001 vs. Control after 24 h. ^#^
*p* < 0.05, ^##^
*p* < 0.01, ^###^
*p* < 0.001 and ^####^
*p* < 0.0001 Control after 48 h.

**Figure 7 biomedicines-11-01643-f007:**
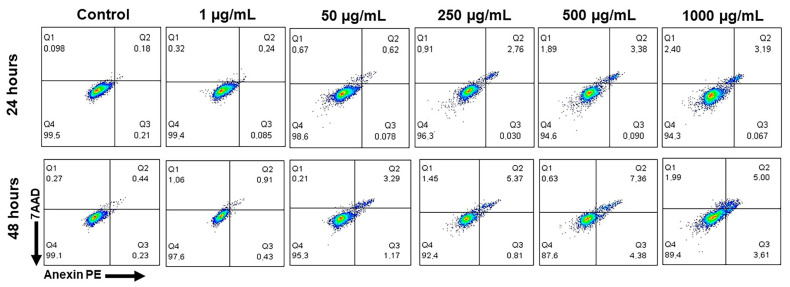
Flow cytometry results for the analysis of cell death due to early and late apoptosis of J774A.1 cells. Dot plot of flow cytometry according to each concentration of NP CB-EDA (1, 50, 250, 500, and 1000 μg/mL). Qualitative data analysis.

**Figure 8 biomedicines-11-01643-f008:**
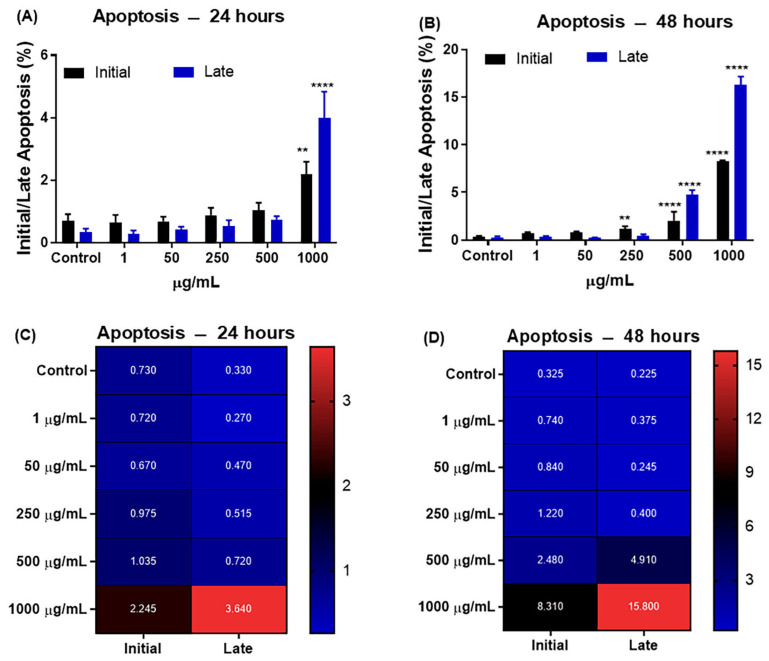
Analysis of percentages of early and late apoptosis via flow cytometry. Cell death due to (**A**) early and (**B**) late apoptosis 24 and 48 h after exposure to CB-EDA NPs. (**C**,**D**) Correlation between early and late apoptosis in the groups analyzed after 24 and 48 h. Blue represents a low percentage of late/early apoptosis expression, whereas red indicates a high percentage of late/early apoptosis expression. ** *p* < 0.01 and **** *p* < 0.0001 vs. Control after 24 and 48 h.

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
