# Peer review of "Carbon Black CB-EDA Nanoparticles in Macrophages: Changes in the Oxidative Stress Pathway and in Apoptosis Signaling"

_biomedicines, 2023, doi:10.3390/biomedicines11061643_

Round 1

Reviewer 1 Report

The manuscript “Carbon black cb-eda nanoparticles in macrophages: changes in the oxidative stress pathway and in apoptosis signaling by Joice et.al., summarizes a proof-of-concept study that demonstrates exploring the cytotoxicity and apoptosis of black carbon functionalized with ethylenediamine nanoparticles on murine cells. The manuscript opens with an introduction about the significance and applications of nanoparticles  materials its properties, mechanism of action, concluding with results. The concept and idea of the work is not novel, because the authors are using a well-known nanoparticle and just checking it on murine cells. This work can be promising with murine cell and might not even show same cytotoxicity with human cancer or other cells for example. Yet, I would request the authors to revisit their manuscript and readdress all their data collected as figures and make visually appealing and easy to understand figures. A lot has been presented in text, but the main idea of it is missing as readers do not have time to spend hours in a manuscript. And in such cases, the manuscript loses its scientific point. I will proceed with a decision once changes are addressed.

However, Kindly revise as below:

1         Line 50, Graphical Abstract, Kindly add the direction arrows to explain better that when nanoparticles interact with the murine cell, ROS/RNS is produced, which leads to apoptosis. This should be presented more artistically.

2         Line 294, figure 1A, kindly acquire a better SEM image of the CB-EDA nanoparticles, based on my understanding these nanoparticles are hexagonal is shape, if so, the SEM image should show that geometry to confirm that shape, else I or any reader cannot understand the credibility of the nanoparticles used was right or not. Purpose of SEM is visual confirmation of shape and size.

3          Line 290, “good particle size” I can only see a blur grey image, how do I know the size is 10 and 40 nm?  figure 1B, a TEM image.  Kindly take good picture, else change the claim, of good particle size and homogeneity.

4         figure 1C, DLS image, I cannot see a graph with Size in mn (x-axis) vs Intensity in a.u (y-axis), that is DLS, what is this?

5         Line 362, figure 4, I like this cell morphology, as visually when I see this picture, I can see that with increase nanoparticle concentration, there is more cell death. But presentation is poor, where is the scale? Please re-arrange it nicely so that cells can be seen.

6         Line 414, figure 7, I like this flow cytometry too, as visually when I see this picture, I can see that with increase nanoparticle concentration, there is more cell death. But presentation is poor, where I can hardly read any text.

7         Kindly upload pictures of the actual cells, taken with microscope

8         How many times was the experiment performed, kindly add the p<0.00x value, probability value, evaluated by t-tests , Annova etc.

9         Introduction: Kindly add a section to how and why this study will be applicable for a journal like biomedicines, more like its applications in that area, and potential uses.

10     Kindly address these few points before discussion of each section? Each of the below listed questions should be the way to address any result or discussion. This way the readers can understand the propose or motto of doing that review rather than presenting your flowcharts.

a.       What is the purpose or goal of this study,  this section , or this sub-section?

b.       What new information does this result give?

c.       What references were used to support the results?

Kindly re-address each section with the above questions. 

11     References. Kindly elaborate more on the introduction sections, and the sub-sections with references. Appropriate references are required to support the claim and results of any review. Include references from near past not 10 yrs ago, science is changing drastically.

Author Response

Response to Reviewer 1 Comments

Point 1: Line 50, Graphical Abstract, Kindly add the direction arrows to explain better that when nanoparticles interact with the murine cell, ROS/RNS is produced, which leads to apoptosis. This should be presented more artistically.

Response 1: We apologize for this error Graphical abstract were revised and corrected.

Point 2: Line 294, figure 1A, kindly acquire a better SEM image of the CB-EDA nanoparticles, based on my understanding these nanoparticles are hexagonal is shape, if so, the SEM image should show that geometry to confirm that shape, else I or any reader cannot understand the credibility of the nanoparticles used was right or not. Purpose of SEM is visual confirmation of shape and size.

Response 2: SEM images were performed and confirmed that the particles do not have hexagonal morphology. One sentence was added in the revised manuscript

Point 3: Line 290, “good particle size” I can only see a blur grey image, how do I know the size is 10 and 40 nm?  figure 1B, a TEM image.  Kindly take good picture, else change the claim, of good particle size and homogeneity.

Response 3: A SEM image was added to improve the visualization of the particles and a sentence in the revised manuscript of the particle size measurement method.

Point 4:  figure 1C, DLS image, I cannot see a graph with Size in mn (x-axis) vs Intensity in a.u (y-axis), that is DLS, what is this?

Response 4: As explained in the text, the 1D image shows the sizes obtained by DLS analysis and their standard deviations in different situations, not the distributions obtained through DLS. To clarify this point, a sentence in the revised manuscript has been rephrased.

Point 5: Line 362, figure 4, I like this cell morphology, as visually when I see this picture, I can see that with increase nanoparticle concentration, there is more cell death. But presentation is poor, where is the scale? Please re-arrange it nicely so that cells can be seen.

Response 5: The image was redone improving its quality and clarifying the doubts pointed out by the referee

Point 6: Line 414, figure 7, I like this flow cytometry too, as visually when I see this picture, I can see that with increase nanoparticle concentration, there is more cell death. But presentation is poor, where I can hardly read any text.

Response 6: The figure 7 was revised and improved

Point 7: Kindly upload pictures of the actual cells, taken with microscope

Response 7: The wells were observed in Axiovert 40 CFL optical microscope (Zeiss), with a 10X objective lens, whose images were captured with the coupled camera model LOD-3000 (Bio Focus) and analyzed by the Future WinJoeTM software version 2.0 in high resolution end of 100X

Point 8:  How many times was the experiment performed, kindly add the p<0.00x value, probability value, evaluated by t-tests , Annova etc.

Response 8: We apologize for the error and rewrite how the statistics part was developed.

Point 9:  Introduction: Kindly add a section to how and why this study will be applicable for a journal like biomedicines, more like its applications in that area, and potential uses.

Response 9: Dear referee, thank you very much for the valuable contributions and suggestions. We made substantial reviews and aggregation of content in the introduction.

Point 10:  Kindly address these few points before discussion of each section? Each of the below listed questions should be the way to address any result or discussion. This way the readers can understand the propose or motto of doing that review rather than presenting your flowcharts.

  1. What is the purpose or goal of this study, this section , or this sub-section?

Response: Objective was described in the introduction section

  1. What new information does this result give?

Response: The results showed that at the highest concentrations the nanoparticle has a cytotoxic potential for human applications

  1. What references were used to support the results?

Response: The results were only described and were referenced only in the discussion according to each methodology

Kindly re-address each section with the above questions. 

Point 11: References. Kindly elaborate more on the introduction sections, and the sub-sections with references. Appropriate references are required to support the claim and results of any review. Include references from near past not 10 yrs ago, science is changing drastically.

Response 11: Dear referee, thank you very much for the valuable contributions and suggestions the references were reviewed and left only those that mention the methodology created by the author himself

Reviewer 2 Report

This manuscript deals with "CARBON BLACK CB-EDA NANOPARTICLES IN MACROPHAGES: CHANGES IN THE OXIDATIVE STRESS PATHWAY AND APOPTOSIS SIGNALING". This article claims that using of CB-EDA could be toxic for for macrophage J774.A1. Study is very interesting and i really enjoyed to  read. Therefore, I suggest a minor correction and require a detailed clarification. Correction to be addressed by the authors as follows: The abstract is not well organized, where the sentences are incomplete and no continuity is there. It would be feasible, if include the significance of the current study in the abstract.

A brief description of how the authors selected information from the literature in the databases, as well as what time period they searched for, is missing.
Authors should justify and expand the information on the advantages of study for biomedical applications.
Authors should specify the main experimental conditions used on the evidences from the literature. Where they briefly describe the most important data reported in the literature in a homogeneous manner and sequence reinforcing the relevance of this work.
Authors should discuss whether the use of antioxidants in functionalization could help to decrease this toxicity?. Also please discuss about the antioxidative role on mitochondria.
Please add below studies to your manuscript in discussion section using below manuscripts:
DOI: 10.1016/j.chemosphere.2022.134826

DOI: 10.1155/2021/2058149 

Conclusions should reaffirm the fundamental contribution of this paper.

Author Response

A  brief description of how the authors selected information from the literature in the databases, as well as what time period they searched for, is missing.
Authors should justify and expand the information on the advantages of study for biomedical applications.
Authors should specify the main experimental conditions used on the evidences from the literature. Where they briefly describe the most important data reported in the literature in a homogeneous manner and sequence reinforcing the relevance of this work.
Authors should discuss whether the use of antioxidants in functionalization could help to decrease this toxicity?. Also please discuss about the antioxidative role on mitochondria.
Please add below studies to your manuscript in discussion section using below manuscripts:
DOI: 10.1016/j.chemosphere.2022.134826

DOI: 10.1155/2021/2058149 

Conclusions should reaffirm the fundamental contribution of this paper.

Response: Thank you very much for the corrections and the suggestions have been changed.

The study was commissioned by the petroleum industry in which it ceded the nanoparticle, so it was not possible to test other functionalization.

Round 2

Reviewer 1 Report

My only review for this version is the SEM image does not show hexagonal nanoparticles, that makes me doubtful to accept it in this form. 

Author Response

Reviewer 1: My only review for this version is the SEM image does not show hexagonal nanoparticles, that makes me doubtful to accept it in this form. 

Response: 

Dear reviewer thank you very much for the corrections. The carbon black nanoparticle (CB) already known in the literature and described as being nanoparticles characterized by being a sphere (VELOSO et al., 2022, COUPETTE, et al, 2021; GUO et al., 2019). Our functionalized nanoparticle with EDA forming CB-EDA is also a sphere according to its sinthesis seen in the article Lima, 2016. What occurs when it is placed in culture medium are clusters reported and demonstrated through the figures in the results of this work.

 VELOSO,W. B;  ALMEIDA, A. T.de F.; RIBEIRO,L. K.,  ASSIS, M.; LONGO, E.;GARCIA, M. A. S.; TANAKA, A. A.;  SILVA, I. S.; DANTAS, L. M. F. Rapid and sensitivity determination of macrolides antibiotics using disposable electrochemical sensor based on Super P carbon black and chitosan composite, Microchemical Journal, Volume 172, Part B, 2022, 106939, ISSN 0026-265X, https://doi.org/10.1016/j.microc.2021.106939.

Coupette F, Zhang L, Kuttich B, Chumakov A, Roth SV, González-García L, Kraus T, Schilling T. Percolation of rigid fractal carbon black aggregates. J Chem Phys. 2021 Sep 28;155(12):124902. doi: 10.1063/5.0058503. PMID: 34598569.

Guo J, Li D, Zhao H, Zou W, Yang Z, Qian Z, Yang S, Yang M, Zhao N, Xu J. Cast-and-Use Super Black Coating Based on Polymer-Derived Hierarchical Porous Carbon Spheres. ACS Appl Mater Interfaces. 2019 May 1;11(17):15945-15951. doi: 10.1021/acsami.9b04779. Epub 2019 Apr 16. PMID: 30942081.

LIMA, M. C. F. S.; AMPARO, S, Z.; RIBEIRO, H.; SOARES, A. L., VIANA, M. M.;  SEARA, L. M.; PANIAGO, R. M.;  SILVA,G. G.; CALIMAN, V. Aqueous suspensions of carbon black with ethylenediamine and polyacrylamide-modified surfaces: Applications for chemically enhanced oil recovery, Carbon, Volume 109, 2016, Pages 290-299, ISSN 0008-6223, https://doi.org/10.1016/j.carbon.2016.08.021